

# Frequency distribution-aware network based on discrete cosine transformation (DCT) for remote sensing image super resolution

Yunsong Li[1] and Debao Yuan[2]

[1] Department of Information Communication, Zhengzhou Electric Power College, Zhengzhou, Henan, China
[2] Department of Geomatics Engineering, China University of Mining and Technology, Beijing, China

## ABSTRACT

Single-image super-resolution technology based on deep learning is widely used in remote sensing. The non-local feature reflects the correlation information between different regions. Most neural networks extract various non-local information of images in the spatial domain but ignore the similarity characteristics of frequency distribution, which limits the performance of the algorithm. To solve this problem, we propose a frequency distribution aware network based on discrete cosine transformation for remote sensing image super-resolution. This network first proposes a frequency-aware module. This module can effectively extract the similarity characteristics of the frequency distribution between different regions by rearranging the frequency feature matrix of the image. A global frequency feature fusion module is also proposed. It can extract the non-local information of feature maps at different scales in the frequency domain with little computational cost. The experiments were on two commonly-used remote sensing datasets. The experimental results show that the proposed algorithm can effectively complete image reconstruction and performs better than some advanced super-resolution algorithms. The code is available at https://github.com/Liyszepc/FDANet.

**Submitted** 10 November 2023
**Accepted** 22 July 2024
**Published** 4 September 2024

Corresponding author
Yunsong Li, liyunsong@zepc.edu.cn

## INTRODUCTION

Remote sensing images have been used in a wide range of applications. However, in remote sensing image acquisition, the image's resolution may be limited by its hardware. Image super-resolution (SR) technology (*Pan et al., 2019*) can utilize low-resolution images to generate high-resolution images, increasing the details of the targets. It can also be used as an image preprocessing method for other remote sensing tasks, such as target recognition (*Ding et al., 2017*), land classification (*Jamil & Bayram, 2018*), target detection (*Wang et al., 2022a*), and so on.

Image SR technology can be divided into multi-image SR (MISR) (*Wang et al., 2018*; *Liu et al., 2022a*) and single-image SR (SISR) methods (*Yu, Li & Liu, 2020*). MISR methods utilize multiple images for image SR, such as hyperspectral image reconstruction tasks (*Li*

*et al., 2023*). SISR methods can utilize a single-image to complete image SR tasks. Thus, SISR technology is more widely used in remote sensing. SISR methods can be divided into three kinds of algorithms, including interpolation-based (*Zhou, Yang & Liao, 2012*), optimization-based (*Tuo et al., 2021*), and learning-based algorithms (*Arun et al., 2020*). The interpolation-based algorithm utilizes the weighted sum of adjacent pixels for image SR, which is fast but has limited performance. It can only handle simple image SR tasks. The optimization-based algorithm utilizes prior knowledge, such as low-rank priori and sparse priori, to complete image reconstruction. The performance of the optimization-based algorithm is better than the interpolation-based algorithm. However, the inference time of the optimization-based algorithm is long, which can not meet the real-time requirements of the tasks. Optimization-based algorithms usually contain multiple hyperparameters, which requires researchers to have more field experience. Learning-based methods contain dictionary learning algorithms (*Wang et al., 2012*; *Gou et al., 2014*; *Ma et al., 2020*) and neural networks (*Tian et al., 2020*). The performance of the dictionary learning method is closely related to the quality of the learned dictionary and requires much computation. Neural networks have been more widely studied in recent years due to their high performance.

Since SRCNN (*Dong et al., 2015*) was proposed in 2015, neural networks have shown superior performance in image SR. Liebe and Korner first introduced SRCNN into remote sensing image (RSI) SR tasks (*Liebel & Korner, 2016*). Furthermore, *Lei, Shi & Zou (2017)* proposed LGCNet for RSI SR, which was a local–global combined network. This network can learn both local and global information about the image simultaneously. *Haut et al. (2019)* introduced the attention mechanism into remote sensing SR tasks, enabling the network to focus more on essential features using the method named RSRCAN. Moreover, a mixed high-order attention network called MHAN (*Chen, Deng & Hu, 2019*) was also proposed for RSI SR. It can mine high-frequency information for image reconstruction. A generative adversarial network (GAN) (*Creswell et al., 2018*) is also widely applied to RSI SR. *Jiang et al. (2019)* proposed EEGAN for image SR, which introduced an edge-enhanced module to the SR network. Moreover, *Lei, Shi & Zou (2019)* also proposed a coupled-discriminate GAN for RSI SR. Recently, the diffusion model has been extensively studied in the SR field due to its high performance. *Xu, Ma & Zhu (2023)* proposed a dual-diffusion model based on dual conditional denoising diffusion probabilistic models for RSI SR, which achieved good results. Previous studies of RSI SR consider the image's local and global information in spatial domain for image reconstruction such as GLORIA (*Wu et al., 2020*), which introduced non local information extraction modules in the network, enabling the algorithm to extract image global features more fully. However, this global information has not been studied in frequency domains. The previous super-resolution methods that utilized frequency information in remote sensing usually only transformed the image into the frequency domain without further extracting non local information from the frequency domain. For remote sensing images, the distribution of frequency information is also an important image feature. The similarity characteristics of frequency distribution between different image regions have not yet been considered. The utilization of this feature can effectively enhance the performance of super-resolution networks.

The frequency distribution feature represents the number of high and low frequency components in the image, while the non-local frequency feature represents the correlation between frequency features in the image.

To develop an effective frequency information extraction network for RSI SR, we propose a frequency distribution aware network (FDANet). This network mainly focuses on the frequency feature of images. FDANet consists of a frequency aware (FA) module and a global frequency (GF) feature fusion module. These two modules focus on extracting feature information from different regions and scales. The FA module can extract the similarity characteristics of frequency distribution between different regions, which has not been considered in previous works. Specifically, we divide the image into multiple patches and perform discrete cosine transformation (DCT) (*Ahmed, Natarajan & Rao, 1974*). After the DCT operation, low-frequency information is located in the upper left corner, and high-frequency information is located in the lower right corner. Then, we combine the frequency weights at the same pixel position in each patch to form new feature maps. Different feature maps represent the distribution of different frequency components. A global feature fusion block is used to extract the information of new features. Finally, we rearrange the pixels and perform inverse discrete cosine transformation (IDCT) to obtain the reconstruction image. Then, we propose the GF module to better extract the non-local information of feature maps at different scales in the frequency domain. We first transform the image into different scales and perform a DCT operation. Subsequently, we also use a global feature fusion block to extract the image's global information, consisting of matrix transpose and convolution operations. It is worth noting that this global information extraction block requires less computation compared to the other global information extraction methods, such as the Transformer structure (*Lu et al., 2022*). The main contributions of this article are summarized as follows:

(1) We propose a frequency distribution aware network for RSI SR. This network can effectively extract the feature information in the frequency domain for image reconstruction.

(2) The FA module is proposed to extract the similarity characteristics of frequency distribution between different regions.

(3) The GF module is proposed to extract the non-local information of feature maps at different scales in the frequency domain.

## RELATED WORK

### Degradation model

Due to hardware device limitations, the RSI resolution cannot meet the practical task's needs. Two degradation factors are considered: noising and downsampling. Thus, the degradation model can be presented as Eq. (1).

$$y = Ax + n, \tag{1}$$

where $A$ denotes the downsampling matrix, $x$ and $y$ represent high-quality and low-quality images, respectively. $n$ is the white Gaussian noise. The function of the SR algorithm is to predict high-quality image $x$ through low-quality image $y$.

## DCT

Proposed algorithm mainly uses DCT operation to extract image information in the frequency domain. DCT is a frequency domain transformation method (*Khayam, 2003*). The two-dimensional DCT operation can be shown as Eq. (2).

$$A(i,j) = \alpha(i)\alpha(j) \sum_{x=0}^{M-1}\sum_{y=0}^{N-1} I(x,y) \cos\left[\frac{\pi(2x+1)i}{2M}\right]\cos\left[\frac{\pi(2y+1)j}{2N}\right],$$

(2)

for $i = 0, 1, 2, \ldots, M-1$ and $j = 0, 1, 2, \ldots, N-1$. $M$ is the number of rows and $N$ is the number of columns. The inverse transformation can be expressed as:

$$I(x,y) = \sum_{i=0}^{M-1}\sum_{j=0}^{N-1} \alpha(i)\alpha(j)A(i,j) \cos\left[\frac{\pi(2x+1)i}{2M}\right]\cos\left[\frac{\pi(2y+1)j}{2N}\right],$$

(3)

and we have:

$$\alpha(i) = \begin{cases} \sqrt{\dfrac{1}{M}}, & i = 0 \\ \sqrt{\dfrac{2}{M}}, & i \neq 0 \end{cases}$$

(4)

$$\alpha(j) = \begin{cases} \sqrt{\dfrac{1}{N}}, & i = 0 \\ \sqrt{\dfrac{2}{N}}, & i \neq 0 \end{cases}.$$

(5)

By performing Eqs. (3) and (4), we can complete the DCT and IDCT operations.

DCT converts image data from the spatial domain to the frequency domain, which can make the features of the image more prominent in the frequency domain, facilitating further feature extraction and processing. DCT coefficients reflect the frequency energy distribution of the image. The low-frequency signal is distributed in the upper left corner of the image after transformation, and the high-frequency signal is in the lower right corner. DCT is often used for image compression. Here, the DCT operation is used to extract the frequency features of the image. Specifically, we extract the features after the DCT operation, including the correlation information of frequency components between different regions and the non-local information of frequency feature maps at different scales. Therefore, two DCT operations complete the extraction of frequency information in different aspects. The mining of frequency information is helpful for the network to make full use of image features for image reconstruction.

## PROPOSED SCHEME

The structure of FDANet is presented based on the degradation model shown in Eq. (1). FDANet consists of two parts, the FA module and the GF module. The FA module mainly learns the frequency distribution features between different regions of the image through DCT operation. The GF module mainly uses DCT to mine the non-local information of the image at different scales in the frequency domain.

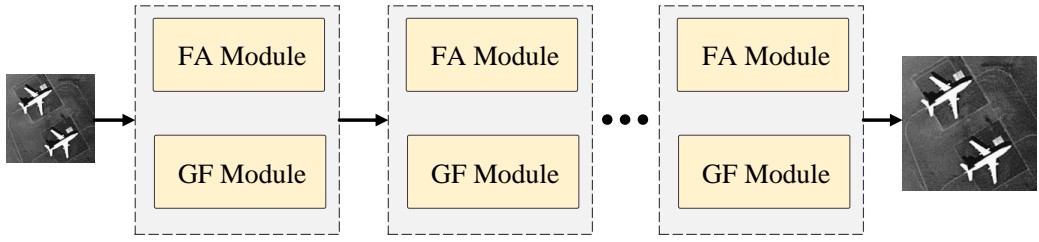

**Figure 1** **The structure of FDANet.**

## FDANet

The structure of FDANet is shown in Fig. 1. Low-resolution images can be reconstructed into high-resolution images through the network. It is worth noting that this network utilizes a recursive neural network strategy to iterate the basic network multiple times, thereby improving its performance and reducing the parameter quantity. Figure 2 depicts a more detailed introduction to the FA and GF modules.

## FA module

In this section, we present the structure of the FA module. The FA module consists of five operations: the DCT operation, rearrange operation, global feature fusion operation, rearrange operation, and IDCT operation. The structure of the FA module is shown in Fig. 3. We first divide the image into $8 \times 8$ pixel patches. Then, we perform a DCT operation for each patch. DCT can reach the energy weight of the low-frequency signal at the upper left corner of the matrix and the energy weight of the high-frequency signal at the lower right corner of the matrix. Therefore, the distribution of different frequency components in the image can be extracted to complete image reconstruction. The image processing flowchart of the FA module is shown in Fig. 4. In other words, the frequency component of each patch can be divided into 64 classes through the DCT operation. The new 64 feature maps represent the distribution of 64 frequency components from low-to-high frequency, which means the new feature maps are composed of the same frequency components in different regions. This arrangement concentrates the frequency component of the image so that the network can better learn the frequency correlation between different patches. Moreover, a global feature fusion block can fully fuse the feature maps of different frequency components.

The structure of the global fusion block is shown in Fig. 5. It is a row-column decoupling strategy. The size of the input image is $b \times n \times h \times w$, where $b$ denotes the batch size, $n$ denotes the channel number, $h$, and $w$ denotes the height and weight of the feature maps. We first perform a convolution operation on the input features. This step will complete information fusion between different feature maps. Then, we transpose the feature maps, and the size of the feature maps is $h \times n \times w$. After the convolution operation, the column pixels are fused. Next, we transpose the feature maps, and the size of the feature maps is $w \times n \times h$. The convolution operation will complete the feature fusion between row pixels. Finally, we transpose the feature map to the original dimension. The non-local information extraction

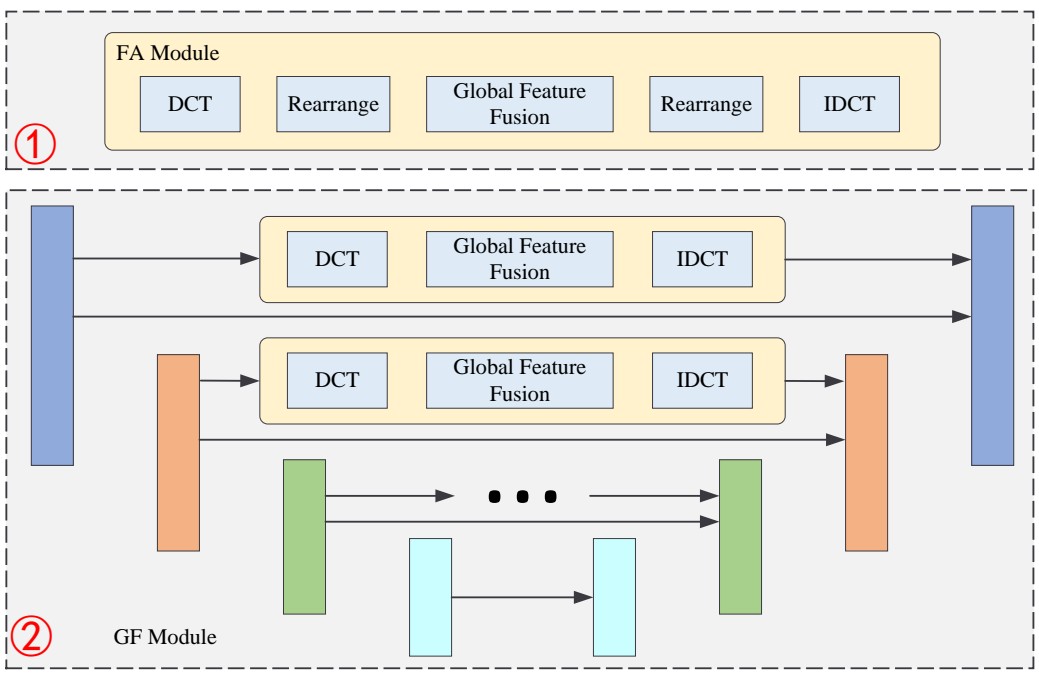

**Figure 2** **The structure of FA module and GF module.** The first part is the FA module, and the second part is GF module.

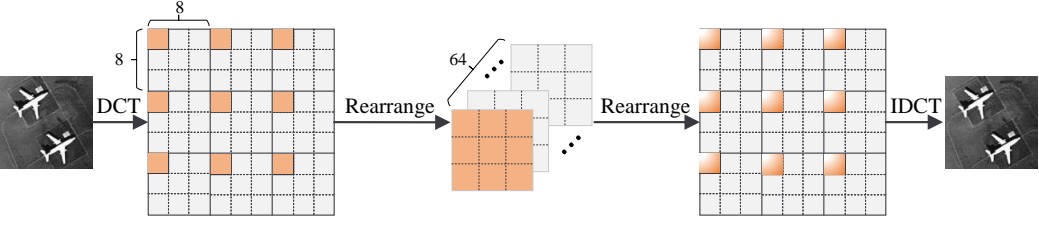

**Figure 3** **The structure of FA module.**

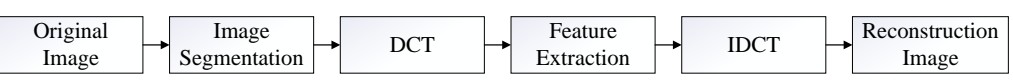

**Figure 4** **FA module image processing flowchart.**

of the row and column pixels is equivalent to the non-local information extraction of all pixels about the image. This process can extract global information from frequency domain feature maps by performing matrix transpose and convolution operations. It requires less computation than the other global information extraction methods.

The information fusion process is shown in Fig. 6. The central pixel is fused with pixels along the $x$-axis, then with pixels along the $z$-axis, and finally with the $y$-axis. It is equivalent

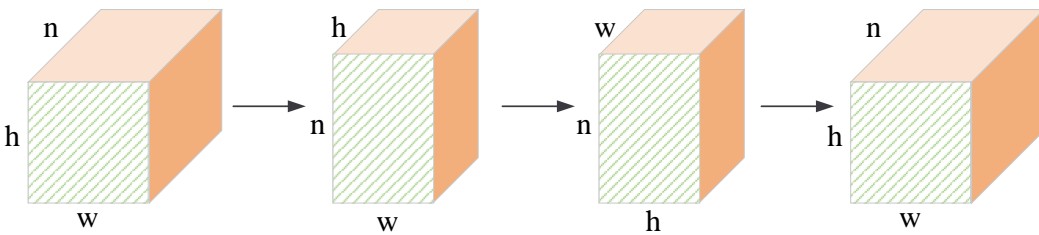

**Figure 5** The structure of global feature fusion block.

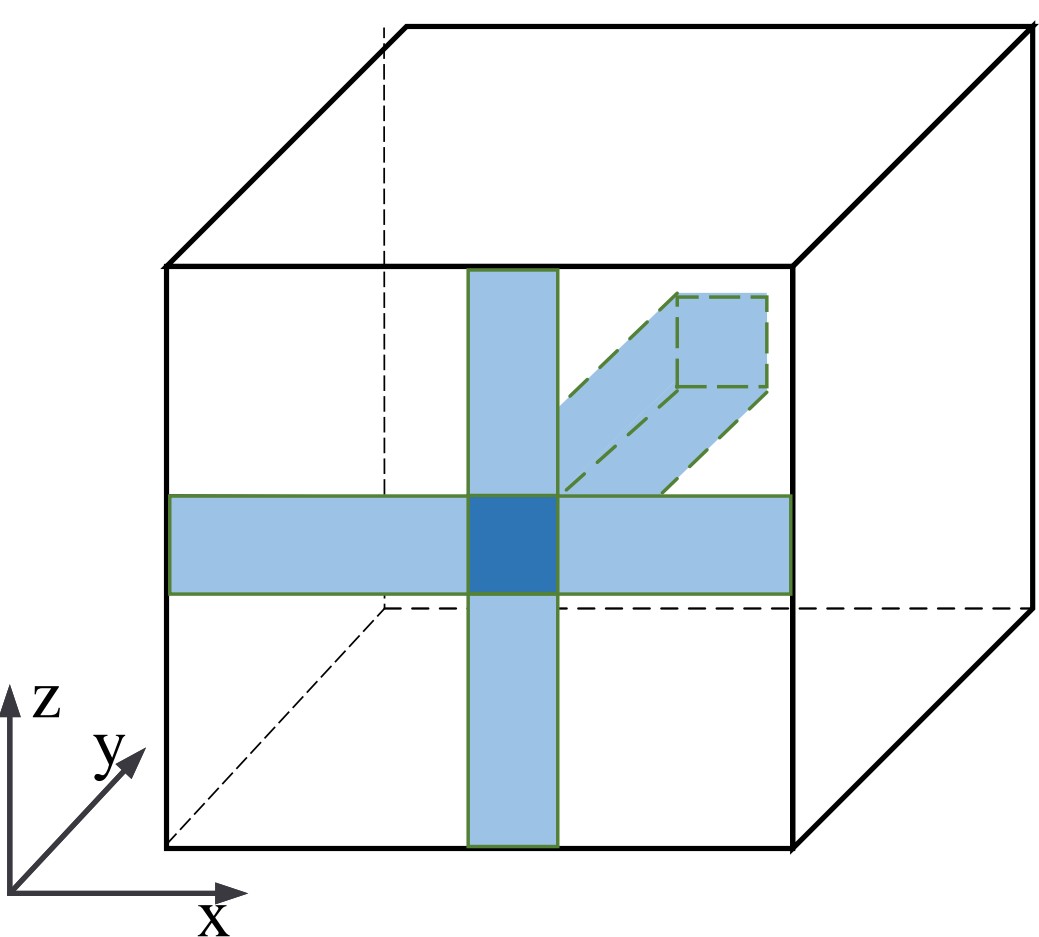

**Figure 6** Schematic diagram of pixel information fusion area.

to fusing the central pixel with all pixels in the feature maps. Our row-column decoupling strategy can extract the global information and avoid directly calculating the correlation information between all pixels. The computational complexity of the network is effectively

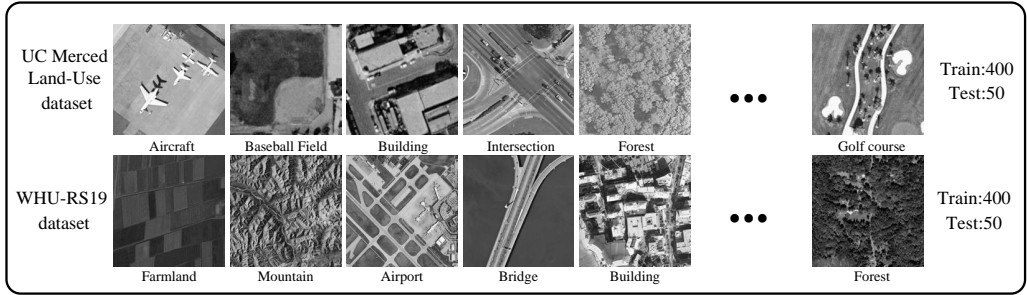

**Figure 7** **Two remote sensing datasets: the UC Merced Land-Use dataset and the WHU-RS19 dataset.**
Images are from the UC Merced Land Use Dataset (http://weegee.vision.ucmerced.edu/datasets/landuse.html) and the the WHU-RS19 dataset (https://paperswithcode.com/dataset/whu-rs1); photographs are in the public domain. *Dai & Yang (2011)*, *Xia et al. (2010)*, *Yang & Newsam (2010)*.

reduced. Finally, the image is rearranged and IDCT operations are performed to obtain the output image.

The FA module rearranges the characteristic information of frequency components so that the network can extract the frequency distribution information more effectively, which facilitates image SR reconstruction.

### GF module

The GF module is shown in the second part of Fig. 2. Its basic structure is Unet. Unet is a classical neural network structure that can extract the multi-scale information of the images; it consists of the encoder and decoder. The structure of Unet has four layers and the size of the feature maps are $b \times n \times h \times w$, $b \times 2n \times \frac{h}{2} \times \frac{w}{2}$, $b \times 4n \times \frac{h}{4} \times \frac{w}{4}$ and $b \times 8n \times \frac{h}{8} \times \frac{w}{8}$, respectively. The feature maps of each layer in the encoder will be transmitted to the corresponding layer of the decoder. This operation can effectively retain the shallow information of the network. We add a frequency information extraction block to the original Unet structure during feature transmission. For example, in the first layer we perform a DCT operation on the feature maps to get the features in the frequency domain. Then, we extract the global features of the image in the frequency domain by global feature fusion block. This global feature fusion block is the same as in the FA module. After the IDCT operation, the features can be transmitted to the decoder.

We perform frequency feature extraction operations on each layer of Unet so that the network can learn the global frequency information of different scales. The feature maps with different scales will contain different frequency components. Therefore, this multi-scale information mining strategy takse advantage of image reconstruction.

## EXPERIMENTS

We perform the experiments on two remote sensing datasets to evaluate the effectiveness of the proposed method (Fig. 7). We have shown several typical scenes. The scenario of the WHU-RS19 dataset is more complex than the UC Merced Land-Use dataset. Therefore, the image reconstruction task under the WHU-RS19 dataset is also difficult.

The UC Merced Land-Use dataset is a typical remote sensing dataset containing 21 kinds of scenes and 100 samples in each class. The pixel resolution of this dataset is 1 foot. The WHU-RS19 dataset is collected from Google Earth by Wuhan University. It contains 19 categories of physical scenes in the satellite imagery, including airports, beaches, bridges, commercial, deserts, rivers, etc. For the convenience of the experiment, we select a total of ten scenes from each dataset. Moreover, we select 40 samples for training and five samples for testing. Ultimately, each dataset has 400 training images and 50 testing images.

The experiments are performed under the Pytorch framework, and are trained on NVIDIA Titan RTX GPUs. Adam is selected as the optimizer. For each training sample, the image is cropped to the patches of $96 \times 96$ pixels as the input. The scale factors are set to 2, 3 and 4. To verify the robustness of the proposed algorithm, the noise levels are set to 0, 3 and 7. We chose additive white Gaussian noise (AWGN) for the experiment. The noise intensity is $\frac{\sigma}{255}$, where $\sigma$ denotes the noise level.

We choose the other five image SR algorithms for comparison, including IMDN (*Hui et al., 2019*), HSENet (*Lei & Shi, 2022*), LGCNet (*Lei, Shi & Zou, 2017*), FENet (*Wang et al., 2022c*) and MAN (*Wang et al., 2022b*). IMDN is a lightweight image SR network with information multi-distillation. LGCNet introduces a local–global combined network for RSI SR. It also utilizes the multi-scale information of the image. HSENet utilizes the self-similarity features of RSIs for image SR. FENet is a feature enhancement network for lightweight RSI SR. MAN is an attention network that consists of a multi-scale large kernel attention structure. To ensure the fairness of the experiment, we make the comparison algorithms have similar computational complexity. The computational complexity of IMDN, HSENet, LGANet, FENet, MAN, and FDANet are 7.65G, 6.27G, 56.23G, 10.76G, 7.92G, and 4.37G, respectively. Here, we use PSNR and SSIM to evaluate the performance of the algorithms. They are two classic image evaluation indices (*Wang et al., 2022c*; *Liu et al., 2022b*). PSNR can be calculated by Eq. (6).

$$PSNR = 20\log_{10} \frac{Max(I_{HR})}{\sqrt{MSE(I_{HR}, G(I_{LR}))}},$$ (6)

where $Max(I_{HR})$ denotes the maximum pixel value in the original image. $G(I_{LR})$ denotes the SR image. MSE represents the mean squared error of the two images. MSE can be expressed as Eq. (7).

$$MSE = \frac{1}{MN} \sum_{i=1}^{N} \sum_{j=1}^{M} (I_{HR}^{ij} - G(I_{LR}^{ij}))^2.$$ (7)

And SSIM can be calculated by Eq. (11).

$$L(\boldsymbol{X}, \boldsymbol{Y}) = \frac{2\mu_x\mu_y + C_1}{\mu_x^2 + \mu_y^2 + C_1}$$ (8)

$$C(\boldsymbol{X}, \boldsymbol{Y}) = \frac{2\sigma_x\sigma_y + C_2}{\sigma_x^2 + \sigma_y^2 + C_2}$$ (9)

$$S(\boldsymbol{X}, \boldsymbol{Y}) = \frac{\sigma_{xy} + C_3}{\sigma_x\sigma_y + C_3}$$ (10)

$$SSIM(\boldsymbol{X}, \boldsymbol{Y}) = L(\boldsymbol{X}, \boldsymbol{Y}) \times C(\boldsymbol{X}, \boldsymbol{Y}) \times S(\boldsymbol{X}, \boldsymbol{Y}) \tag{11}$$

where $\mu_x$ and $\mu_y$ denote the average values of pixels in image $X$ and $Y$, respectively. $\sigma_x$ and $\sigma_y$ denote the variance of image $X$ and $Y$. $\sigma_{xy}$ represents the covariance of image $X$ and $Y$. $C_1$, $C_2$ and $C_3$ are constants.

We first perform the experiments on two remote sensing datasets. Then, several ablation experiments are performed to further demonstrate the proposed module's contribution to improving the algorithm's performance.

### Remote sensing dataset1

The algorithms were applied to the UC Merced LandUse dataset. The experiment results are shown in Table 1.

FDANet performs best. HSENet achieves the second-best results, which may be due to the algorithm's utilization of image multi-scale self-similarity features. We average the algorithm's results for image reconstruction under different conditions, including multiple scale factors and noise levels. The average PSNR values of IMDN, HSENet, LGCNet, FENet, MAN, and FDANet are 26.89 dB, 27.56 dB, 27.41 dB, 27.55 dB, 27.28 dB, and 27.78 dB, respectively. The average SSIM values of IMDN, HSENet, LGCNet, FENet, MAN, and FDANet are 0.724, 0.744, 0.748, 0.734, 0.749, and 0.775, respectively. Table 1 shows that FDANet outperforms the other competitive methods under various noise levels and scaling factors, which shows the algorithm's effectiveness.

### Remote sensing dataset2

The experimental results on the WHU-RS19 dataset are shown in Table 2. The WHU-RS19 dataset is more complex, thus its average PSNR and SSIM values are lower than the UC Merced Land-Use dataset. The proposed algorithm still performs best. HSENet has the second-best performance. The average PSNR values of IMDN, HSENet, LGCNet, FENet, MAN, and FDANet are 25.97 dB, 26.54 dB, 26.41 dB, 26.53 dB, 26.20 dB, and 26.59 dB, respectively. The average SSIM values of IMDN, HSENet, LGCNet, FENet, MAN, and FDANet are 0.675, 0.702, 0.705, 0.676, 0.695, and 0.731, respectively. Table 2 shows that FDANet performs well in more complex remote sensing datasets, which verifies the algorithm's robustness.

### Visual results

In this section, we present the visual results for algorithm comparison. We randomly selected several scale factors and noise levels to demonstrate the effectiveness of the algorithm in various conditions. Figure 8 is the aircraft scene with a scale factor of 2 and noise level of 0. LR and HR denote low-resolution and high-resolution images, respectively. The low-resolution image is obtained by the nearest neighbor interpolation method. The large image on the left is the whole HR image. We enlarge the area in the red box for algorithm comparison. The large image on the left is the whole HR image. The low-resolution image has severe sawtooth effects. The edge information of the image is lost. Multiple SR algorithms can restore image details to varying degrees. The edges of FDANet

**Table 1  Scores of PSNR and SSIM for different SR algorithms with remote sensing dataset 1.** The best results are shown in bold, and the second-best results are underlined.

| Dataset | Scale | Noise | Metrics | IMDN (2019) | HSENet (2022) | LGCNet (2017) | FENet (2022) | MAN (2022) | FDANet |
|---------|-------|-------|---------|-------------|---------------|---------------|--------------|------------|--------|
| | x2 | 0 | PSNR | 30.54 | 31.21 | 31.15 | 31.01 | 31.01 | **31.38** |
| | | | SSIM | 0.841 | 0.859 | 0.855 | 0.839 | 0.866 | **0.879** |
| | x3 | 0 | PSNR | 26.68 | 27.45 | 27.30 | 27.42 | 27.17 | **27.83** |
| UC Merced | | | SSIM | 0.723 | 0.751 | 0.756 | 0.740 | 0.753 | **0.784** |
| Land-Use | x3 | 3 | PSNR | 26.56 | 27.23 | 27.16 | 27.24 | 26.99 | **27.61** |
| dataset | | | SSIM | 0.716 | 0.739 | 0.748 | 0.732 | 0.744 | **0.771** |
| | x3 | 7 | PSNR | 26.19 | 26.53 | 26.69 | 26.68 | 26.49 | **27.01** |
| | | | SSIM | 0.698 | 0.701 | 0.724 | 0.703 | 0.718 | **0.740** |
| | x4 | 0 | PSNR | 24.49 | 25.40 | 24.73 | **25.42** | 24.73 | 25.08 |
| | | | SSIM | 0.642 | 0.671 | 0.657 | 0.658 | 0.664 | **0.699** |

**Table 2  Scores of PSNR and SSIM for different SR algorithms with remote sensing dataset 2.** The best results are shown in bold, and the second-best results are underlined.

| Dataset | Scale | Noise | Metrics | IMDN (2019) | HSENet (2022) | LGCNet (2017) | FENet (2022) | MAN (2022) | FDANet |
|---------|-------|-------|---------|-------------|---------------|---------------|--------------|------------|--------|
| | x2 | 0 | PSNR | 28.62 | 28.97 | 29.13 | 28.91 | 28.89 | **29.18** |
| | | | SSIM | 0.798 | 0.828 | 0.821 | 0.788 | 0.821 | **0.842** |
| | x3 | 0 | PSNR | 25.80 | **26.71** | 26.30 | 26.52 | 26.10 | 26.60 |
| | | | SSIM | 0.671 | 0.705 | 0.709 | 0.677 | 0.694 | **0.739** |
| WHU-RS19 | x3 | 3 | PSNR | 25.69 | 26.40 | 26.16 | 26.32 | 25.96 | **26.42** |
| dataset | | | SSIM | 0.665 | 0.694 | 0.701 | 0.670 | 0.687 | **0.728** |
| | x3 | 7 | PSNR | 25.38 | 25.91 | 25.76 | 25.76 | 25.60 | **26.00** |
| | | | SSIM | 0.650 | 0.661 | 0.680 | 0.645 | 0.667 | **0.702** |
| | x4 | 0 | PSNR | 24.35 | 24.69 | 24.68 | 25.17 | 24.47 | **24.76** |
| | | | SSIM | 0.591 | 0.623 | 0.614 | 0.600 | 0.606 | **0.645** |

are clearer than the other algorithms. HSENet and FDANet have similar performance. The performance of IMDN is limited. Therefore, FDANet performs best.

Figure 9 shows an intersection scene with a scale factor of 3 and noise level of 0. The task of the three times SR is more challenging than two times SR task. LR image will lose more details. SR algorithms can effectively reconstruct image details. Both FDANet and MAN have good image reconstruction results. FDANet can fully mine the image's frequency and spatial domain information, so it can effectively reconstruct the image details and has the best performance.

Figure 10 shows a baseball field scene with a scale factor of 3 and noise level of 7. This is the most challenging task. The edge of the LR image is no longer clear. All SR algorithms can recover partial edge information. Moreover, FDANet has the best reconstruction results. The edge of the image is sharper, and noise removal is more effective.

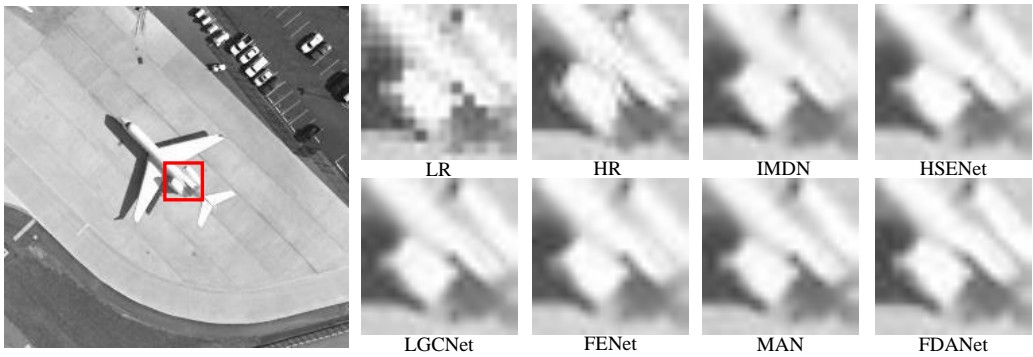

**Figure 8  Aircraft (UC Merced Land-Use dataset).** The scale factor is 2 and the noise level is 0. Images are from the UC Merced Land Use Dataset (http://weegee.vision.ucmerced.edu/datasets/landuse.html) and in the public domain).

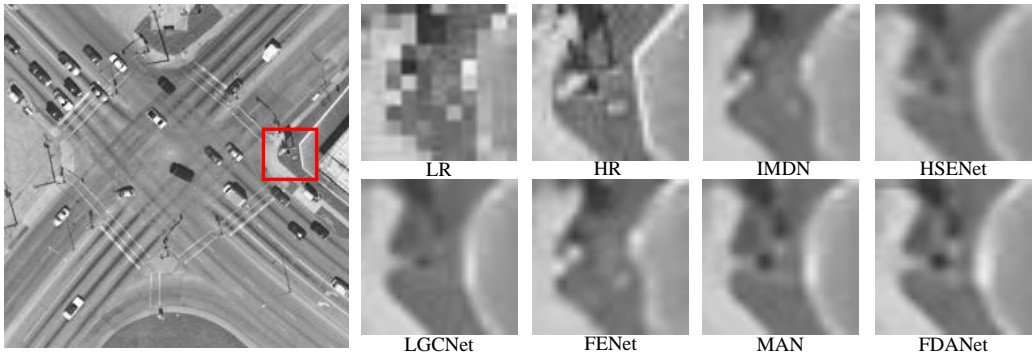

**Figure 9  Intersection (WHU-RS19 dataset).** The scale factor is 3 and the noise level is 0. Images are from the WHU-RS19 dataset (https://paperswithcode.com/dataset/whu-rs1) and in the public domain.

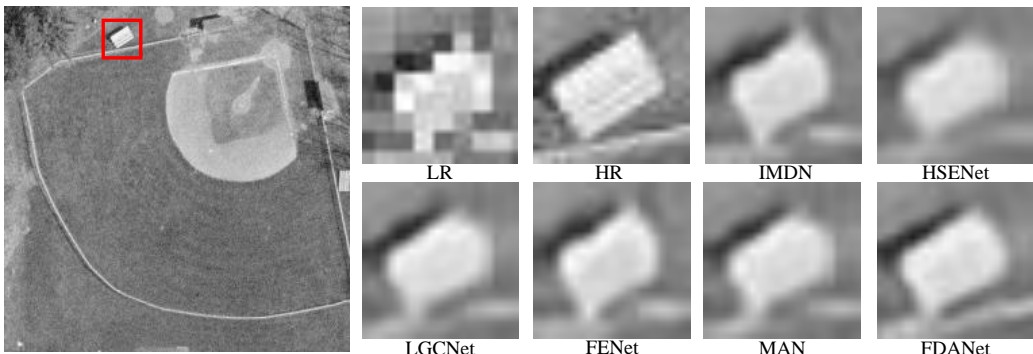

**Figure 10  Baseball field (UC Merced Land-Use dataset).** The scale factor is 3 and the noise level is 7. Images are from the UC Merced Land Use Dataset (http://weegee.vision.ucmerced.edu/datasets/landuse.html) and in the public domain.

**Table 3  Ablation experiments.** The best results are shown in bold.

| Model | Scale | PSNR1 | SSIM1 | PSNR2 | SSIM2 |
|---|---|---|---|---|---|
| | x2 | 30.92 | 0.861 | 28.78 | 0.821 |
| Base model | x3 | 26.39 | 0730 | 25.81 | 0.715 |
| | x4 | 23.90 | 0.635 | 24.67 | 0.621 |
| | x2 | 31.27 | 0.883 | 29.18 | 0.842 |
| Base model+FA | x3 | 27.27 | 0.774 | 26.20 | 0.735 |
| | x4 | 24.05 | 0.643 | **24.78** | **0.648** |
| | x2 | 31.09 | 0.874 | 28.85 | 0.831 |
| GF | x3 | 26.44 | 0.737 | 26.01 | 0.720 |
| | x4 | 24.67 | 0.655 | 24.63 | 0.622 |
| | x2 | **31.38** | **0.879** | **29.18** | **0.842** |
| FDANet | x3 | **27.83** | **0.784** | **26.60** | **0.739** |
| | x4 | **25.08** | **0.699** | 24.76 | 0.645 |

In summary, the proposed algorithm has a good visual reconstruction result. The algorithm is robust to different scale factors and different noise levels. Therefore, FDANet can effectively reconstruct image details and edges.

## Ablation experiments

In this section, we will perform the ablation experiments. The ablation experiment results are shown in Table 3. PSNR1 and PSNR2 indicate that the experiments are conducted under the UC Merced LandUse dataset and the WHU-RS19 dataset. This article has two innovation modules: the FA and GF modules. Thus, we perform four models for comparison. The backbone of the base model is Unet., which was first proposed in 2015 (*Ronneberger, Fischer & Brox, 2015*). The base model with the FA module denotes the FDANet model without the gobal frequency feature fusion blocks in the GF module. The GF model denotes the FDANet model without the FA module. The structure of the base model with the FA module is shown in Fig. 11, and the structure of the GF model is shown in Fig. 12. As shown in Fig. 11, the upper half of Fig. 11 is the FA module and the lower half is the base model. The comparison between Fig. 11 and the basic model can demonstrate the effectiveness of the FA module. Blocks of the same color represent Unet feature maps of the same scale. Skip connections were used for feature concatenation between the same scales. The global feature fusion module is specifically introduced in Fig. 6. The global feature fusion module was used during the skip connection process on the basis of Unet, which consists of GF model (Fig. 12). The comparison between Fig. 12 and the basic model can demonstrate the effectiveness of the GF module. Finally, the FA model can extract the similarity characteristics of frequency distribution between different regions and GF model can extract the non-local information of feature maps at different scales in the frequency domain.

Comparing the base model alone and the base model with the FA module proves the validity of the FA module. This frequency-aware strategy is effective. Comparing the base model and the GF module proves the validity of the global frequency feature

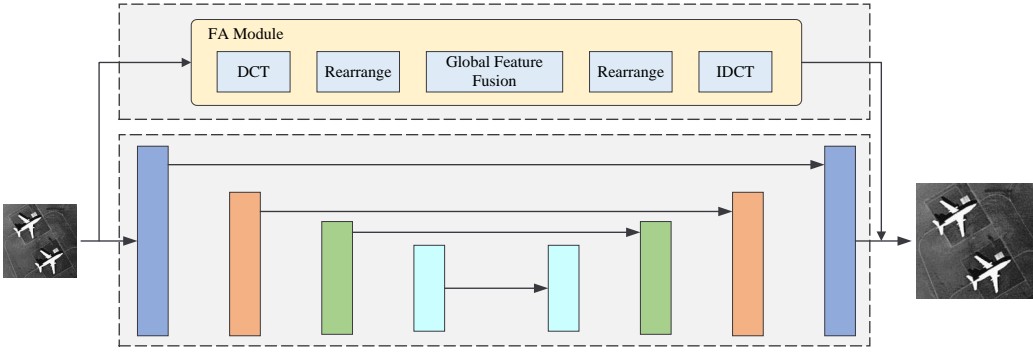

**Figure 11  Base model with FA module.**

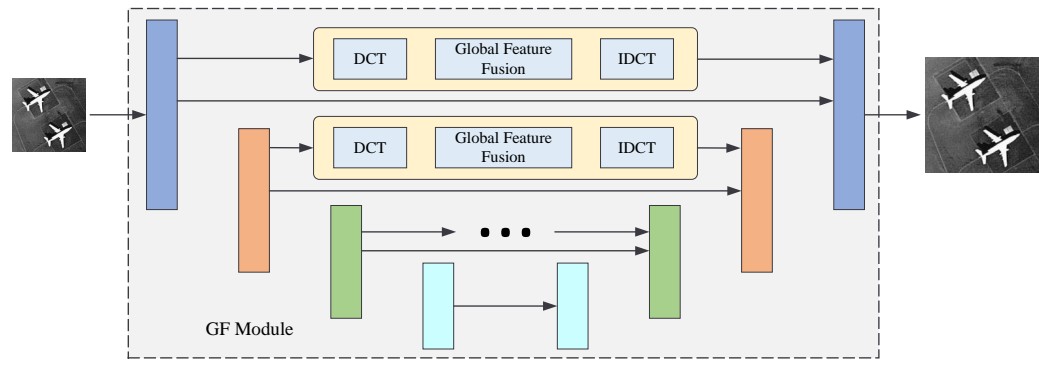

**Figure 12  GF model.**

fusion strategy. This extraction of the global frequency feature is beneficial to image reconstruction. Therefore, the two proposed modules are both effective. FDANet achieves the best performance.

## CONCLUSION

This article proposes a frequency distribution aware network for RSI SR. The network utilizes the FA and GF modules to extract frequency information from images. The FA module extracts the similarity characteristics of frequency distribution between different regions. The GF module effectively extracts the non-local information of feature maps at different scales in the frequency domain. Therefore, FDANet may fully utilize the frequency domain information of the image for image reconstruction. We conducted multiple experiments on two remote sensing datasets to verify the effectiveness of the algorithm under different scale factors and noise levels. The proposed method provides two information perspectives that can be used for image super-resolution reconstruction, which are the frequency distribution characteristics and non local frequency characteristics. The utilization of these two types of information can compensate for the shortcomings of traditional spatial convolutional neural networks in extracting image features for image

super-resolution. However, the proposed method does not consider more complex image degradation issues, such as image blurring. The blurring factor can significantly change the frequency distribution information of the image. Therefore, whether the proposed method is still applicable is an aspect that needs to be studied in the future. Based on the proposed algorithm, the future research directions can be considered:

(1) Unsupervised methods can effectively utilize unlabeled data, therefore future work will use unsupervised networks for remote sensing image processing.
(2) Future work will consider more complex degradation factors in RSI SR tasks, such as blurring kernels.

### Funding
This research was supported by the National Natural Science Foundation of China, grant number 52174160. The funders had no role in study design, data collection and analysis, decision to publish, or preparation of the manuscript.

### Grant Disclosures
The following grant information was disclosed by the authors:
National Natural Science Foundation of China: 52174160.

### Competing Interests
The authors declare there are no competing interests.

### Author Contributions
- Yunsong Li conceived and designed the experiments, performed the experiments, analyzed the data, performed the computation work, prepared figures and/or tables, and approved the final draft.
- Debao Yuan conceived and designed the experiments, performed the experiments, analyzed the data, authored or reviewed drafts of the article, and approved the final draft.

### Data Availability
Data are available at Zenodo:

Li, Y. (2024). Frequency distribution-aware network based on DCT for remote sensing image super resolution. Zenodo. https://doi.org/10.5281/zenodo.10954530

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
