# Peer review of "Frequency distribution-aware network based on discrete cosine transformation (DCT) for remote sensing image super resolution"

_PeerJ Computer Science, doi:10.7717/peerj-cs.2255_

## Round 0.1 · original submission · Major Revisions

Based on the reviewers' comments, please revise the manuscript accordingly.

**Language Note:** PeerJ staff have identified that the English language needs to be improved. When you prepare your next revision, please either (i) have a colleague who is proficient in English and familiar with the subject matter review your manuscript, or (ii) contact a professional editing service to review your manuscript. PeerJ can provide language editing services - you can contact us at [email protected] for pricing (be sure to provide your manuscript number and title). – PeerJ Staff

Reviewer 1 ·

Basic reporting

The manuscript presents two innovative approaches for enhancing the resolution of satellite images: the frequency-aware module and the global frequency feature fusion module. The evaluation methods employed demonstrate promising outcomes. However, I have several comments and suggestions for improvement:
In the introduction, the authors reviewed technologies implemented to increase the resolution of remote sensing images. They mentioned that "In the previous works of RSI SR, many researchers consider the image’s local and global information, frequency, and time domain information for image reconstruction." But what are these studies? Why these factors are helpful in image reconstructin? What is the difference between the proposed frequency distribution and previous studies which also took advantage of frequency and time domain? Why is it necessary to design another specific algorithm to increase resolution for satellite image without adopting existing resolution enhancement techniques for any image? By answers these questions will the authors help the readers to understand the motivation behind this study.

Experimental design

In section 2.2 DCT , the article mentions the time-frequency domain transformation method. Is it a time related algorithm? If so, how is time factor represeted in the equations?

What are PSNR and SSIM short for? Why are they suitable for evaluating model performance? It would also be helpful to include further information on these two methods, suich as whether larger values mean a good model result.

In section 4.3 and Figure 7, the article shows both the LR and HR images. To clarify, is the larger aircraft image a LR image? Was the proposed algorithm applied to the LR or the HR image?

It seems that the values applied for parameters in each sample test are different. For instance, scale factor is 2 and noise level is 0 for Figure 7; those are 3 and 0 for Figure 8, and 3 and 7 for Figure 9. Is there any specific reason of choosing those values? How to systematically decide those parameters when applying the proposed algorithm for other remote sensing images?

Need more details to understand Figures 10 and 11, including the colored blocks, arrows, and the difference of FA and FG models. What is the base model? What is the association and difference between Figure 4 and Figure 10? It seems both describe the FA model.

Need further description on what is ablation experiments, and why they are adopted in the study. To my understanding, ablation study is to remove parts of the system so as to get a better evaluation of the overall model performance. It seems this study does not include enough details of the experiment.

Validity of the findings

I would recommend the authors to extend the conclusion with more practical applications rather than a brief summary of the results. How will your algorithm help in industry and academia? What is the importance of the proposed method? Is your algorithm only benefitial to increase the resolution of remote sensing images or any low-resolution images in general?

I am also curious what is the average processing time of each image by using the proposed method compared with other method since increasing image resolution should be a balance of image quality and processing time.

Additional comments

- Clarify the definition of variables (n, h, w) at their first mention in the text on line 146, instead of mentioning them in the next section on line 167.

- Need definitions and further explanations for key concepts, such as local inforamtion, global information, frequency, and frequency distribution.

- Explain the meaning of underscored values in Tables 1 and 2. Are they indicative of second-best results?

- Clarify the difference between PSNR1 and PSNR2 in Table 3.

- Correct formatting inconsistencies, such as removing the space in "frequency- aware" in the abstract, and adding a hyphen in "single image" on line 37 so as to be consistent with "multi-image" on line 36.

Reviewer 2 ·

Basic reporting

1. The manuscript introduces two innovative methods for enhancing satellite image resolution: the frequency-aware module and the global frequency feature fusion module. While the evaluation shows promising results, there are comments and suggestions for improvement.

2. In the introduction, the authors discuss technologies used to enhance remote sensing image resolution. However, details about prior studies, the importance of specific factors in image reconstruction, distinctions from the proposed frequency distribution, and the need for a new algorithm exclusively for satellite images remain unclear. Addressing these points would clarify the study's motivation for readers.

Experimental design

1. Section 2.2 mentions the time-frequency domain transformation method. Clarification is needed on whether it's time-related and how time is represented in the equations.

2. What do PSNR and SSIM stand for, and why are they suitable for evaluating model performance? Clarify if larger values indicate better results.

3. In section 4.3, Figure 7 displays LR and HR images. Is the larger aircraft image LR, and was the proposed algorithm applied to LR or HR?

4. Parameter values vary between sample tests. Why were specific values chosen, and how are parameters decided for other images?

5. Some figures lack details on colored blocks, arrows, and the difference between FA and FG models. Clarify the base model and the relationship between figures like 4 and 10.

6. More information is needed on ablation experiments and why they are adopted. The study lacks sufficient experimental details.

Validity of the findings

1. I suggest expanding the conclusion to discuss practical applications and the significance of the proposed algorithm in industry and academia. Clarify how the method benefits beyond remote sensing images and address the average processing time compared to other methods, considering the balance between image quality and processing time.

Additional comments

1. Provide the definition of variables (n, h, w) when first mentioned in the text on line 146, rather than delaying it until the next section on line 167.

2. Include definitions and further explanations for key concepts.

---

## Round 0.2 · Minor Revisions

Dear authors,

Thank you for the revised manuscript. There was a problem with the re-review by Reviewer #2 and so two new reviewers (Reviewer #3 and Reviewer #4) were sought for the review process. Based on the reviewers' comments, minor revision is required for the manuscript. When submitting the revised version of your article, it will be better to also address the following:

1. Explanation of the equations should be checked. Please use equation numbers for referencing the equations. Do not use "as", “following” “as follows”, etc. Furthermore, appropriate references should be used for relevant equations. They seem they are firstly used in this paper.
2. Pros and cons of the methods should be clarified. What are the limitation(s) methodology(ies) adopted in this work? Please indicate practical advantages, and discuss research limitations.

Best wishes,

Reviewer 1 ·

Basic reporting

Thanks for revising the article. All my research-related questions have been answered and the language is improved.

Experimental design

Satisfied.

Validity of the findings

Satisfied.

·

Basic reporting

This article proposes a frequency distribution perception network based on DCT for the field of remote sensing satellite image super-resolution, and proposes two innovative modules to improve the resolution of remote sensing images: frequency perception module and global frequency feature fusion module, which focus on different regions and scales of the image, respectively. The displayed evaluation indicators show satisfactory results.

Experimental design

The author's response provided a good response to the reviewer's comments, including an explanation of evaluation indicators, which helped the reviewer have a clearer understanding of the field of super-resolution reconstruction.
But when the author introduced the DCT method, they only listed the formulas without elaborating on the principles. It is recommended to supplement the pseudocode or flowchart of the algorithm.

Validity of the findings

The author's response was very clear and comprehensive, resolving the reviewer's doubts.I suggest that the models in Tables 1 and 2 can be marked with references and years so that readers can read them more clearly.

Additional comments

The reference link of the appendix dataset is not the original link of the dataset, please update it.

Reviewer 4 ·

Basic reporting

no comment

Experimental design

In the experimental section, this paper provides a detailed introduction to the evaluation metrics of the experimental results and the datasets used. It also compares the proposed model with other algorithms and conducts ablation experiments to effectively validate the superiority of the proposed model.

Validity of the findings

In terms of numerical results, the method proposed in the paper indeed shows improvements compared to other methods.
However, the visual effects are not significantly noticeable. It is hoped that the author can make improvements so that the advantages of the proposed method are also reflected in the visual effects.

---

## Round 0.3 · accepted · Accept

Dear authors,

Thank you for the revision and for clearly addressing all the reviewers' comments. Your paper now seems ready for publication in light of this revision.

Best wishes,

·

Basic reporting

All my research-related questions have been answered.

Experimental design

After modifications, the experiment results was clearer and the problem was solved.

Validity of the findings

The model has some value.

Reviewer 4 ·

Basic reporting

no comment

Experimental design

no comment

Validity of the findings

no comment

Additional comments

The manuscript authors have made thorough revisions according to the reviewer's comments. I have no further questions and think this manuscript is acceptable.